# Omega-6 Polyunsaturated Fatty Acids Enhance Tumor Aggressiveness in Experimental Lung Cancer Model: Important Role of Oxylipins

**DOI:** 10.3390/ijms23116179

**Published:** 2022-05-31

**Authors:** Mayra Montecillo-Aguado, Belen Tirado-Rodriguez, Gabriela Antonio-Andres, Mario Morales-Martinez, Zhen Tong, Jun Yang, Bruce D. Hammock, Rogelio Hernandez-Pando, Sara Huerta-Yepez

**Affiliations:** 1Programa de Doctorado en Ciencias Biomédicas, Facultad de Medicina, Universidad Nacional Autonoma de Mexico (UNAM), Mexico City 04510, Mexico; mayramontecillo@hotmail.es; 2Unidad de Investigacion en Enfermedades Oncologicas, Hospital Infantil de Mexico, Federico Gomez, Mexico City 06720, Mexico; bely_16@hotmail.com (B.T.-R.); gabya_24@yahoo.com.mx (G.A.-A.); ixnergal@gmail.com (M.M.-M.); 3Molecular Toxicology Interdepartmental Program and Environmental Health Sciences, University of California, Los Angeles, CA 90095, USA; ztong@coh.org; 4Department of Pathology & Laboratory Medicine, University of California, Los Angeles, CA 90095, USA; 5Comprehensive Cancer Center, Department of Entomology and Nematology, University of California, Davis, CA 95616, USA; junyang@ucdavis.edu (J.Y.); bdhammock@ucdavis.edu (B.D.H.); 6Experimental Pathology Section, Department of Pathology, National Institute of Medical Science and Nutrition, Salvador Zubiran (INCNSZ), Mexico City 14080, Mexico; rhdezpando@hotmail.com

**Keywords:** polyunsaturated fatty acids (PUFAs), omega-3 (ω-3) PUFAs, omega-6 (ω-6) PUFAs, oxylipins, tumor growth, aggressively

## Abstract

Lung cancer is currently the leading cause of cancer death worldwide; it is often diagnosed at an advanced stage and bears poor prognosis. It has been shown that diet is an important environmental factor that contributes to the risk and mortality of several types of cancers. Intake of ω-3 and ω-6 PUFAs plays an important role in cancer risk and progression. Current Western populations have high consumption of ω-6 PUFAs with a ratio of ω-6/ω-3 PUFAs at 15:1 to 16.7:1 This high consumption of ω-6 PUFAs is related to increased cancer risk and progression. However, whether a diet rich in ω-6 PUFAs can contribute to tumor aggressiveness has not been well investigated. We used a murine model of pulmonary squamous cell carcinoma to study the aggressiveness of tumors in mice fed with a diet rich in ω-6 PUFAs and its relationship with oxylipins. Our results shown that the mice fed a diet rich in ω-6 showed a marked increase in proliferation, angiogenesis and pro-inflammatory markers and decreased expression of pro-apoptotic proteins in their tumors. Oxylipin profiling revealed an upregulation of various pro-tumoral oxylipins including PGs, HETEs, DiHETrEs and HODEs. These results demonstrate for the first time that high intake of ω-6 PUFAs in the diet enhances the malignancy of tumor cells by histological changes on tumor dedifferentiation and increases cell proliferation, angiogenesis, pro-inflammatory oxylipins and molecular aggressiveness targets such as NF-κB p65, YY1, COX-2 and TGF-β.

## 1. Introduction

Lung cancer is currently the most common cancer worldwide in terms of death rate. The majority of malignant neoplasms of the lung (85%) are non-small cell lung carcinomas (NSCLC), [1,2,3]. In spite of improvements in detection and standard of care, lung cancers are often diagnosed at an advanced stage and bear poor prognosis [4]. Although the combination of surgery, chemotherapy and immunotherapy can improve survival, the prognosis of lung cancer is still limited [5,6]. Diet is recognized as an important environmental factor that contributes to the risk and mortality of several types of cancers [7], especially polyunsaturated fatty acids (PUFAs) of omega-3 (ω-3) and omega-6 (ω-6). The recommended dietary ratio of ω-6/ω-3 PUFAs for health benefits is 1.1–2:1. However, the current Western population has a high consumption of ω-6 PUFAs with the ratio of ω-6/ω-3 PUFAs at 15:1 to 16.7:1 [8].

There is increasing evidence that PUFAs play a role in cancer risk and progression. PUFAs can be classified into different categories with a variety of biological functions. The ω-3 family of PUFAs includes alpha-linolenic acid (ALA), eicosapentaenoic acid (EPA), and docosahexaenoic acid (DHA), while the ω-6 family includes linolenic acid (LA) and arachidonic acid (ARA). PUFAs, through oxidative reactions, generate oxylipins (lipid derivates that are the main mediators of the effects of PUFAs) by the cyclooxygenase pathway (COX), the lipoxygenase pathway (LOX), and the cytochrome P450 pathway (CYP450) [9]. EPA and DHA are precursors for anti-inflammatory oxylipins while LA and ARA are precursors for pro-inflammatory oxylipins [10]. Collectively, PUFAs play crucial roles in maintaining cellular homeostasis, and perturbations in dietary intake or PUFAs metabolism could result in cellular dysfunction and contribute to cancer risk and progression [11]. In addition, several studies have demonstrated that ω-6 PUFAs induce progression in certain types of cancer [12,13].

There is much clinical evidence indicating the effect of ω-6 PUFAs on lung cancer progression. For example, Liu et al. showed that patients with lung adenocarcinoma and squamous cell carcinoma presented high levels of free fatty acids ARA and LA and their hydroxyeicosatetraenoic acids (HETE) metabolites compared with control subjects without cancer and they propose these as possible markers [14]. Regarding experimental models, Panigrahy et al. showed that epoxyeicosatrienoic acids (EETs) stimulated primary tumor growth, multiorgan metastasis, and escape from tumor dormancy [15]. In contrast, Zhang et al. demonstrated that 16,17-EpDPE and 19,20-EpDPE plus t-AUCB (inhibitor of soluble epoxide hydrolase) decrease LLC metastasis in a murine model [16]. In addition, in an in vitro study, Ling et al. reported that the inhibition of ARA metabolism by a natural triterpenoid, pachymic acid (PA) in human NSCLC A549 cells produced growth inhibition, apoptosis and disruption of mitochondrial membrane potential, in part through its inhibition of the MAPKs and NF-κB signaling pathways [17]. Furthermore, Xia et al. showed that reduction in the ω-6/ω-3 PUFAs ratio in A549 cells decreased their invasive potential by downregulation of several adhesion-/invasion-related genes (MMP-1, integrin-alpha2 and nm23-H4) [18]. Moreover, Yu et al. reported that overexpression of CYP450 ω-hydroxylase CYP4A11 increase 20-HETE levels, promoting angiogenesis and metastasis by upregulation of VEGF and MMP9 in A549 cells [19]. 

Our group have demonstrated that administration of TCDD (2,3,7,8-Tetrachlorodibenzo-ρ-dioxin), which is a ligand of the aryl hydrocarbon receptor (AHR), plus TPPU (1-(1-Propionylpiperidin-4-yl)-3-(4-(trifluoromethoxy)phenyl) urea—an inhibitor of soluble epoxide hydrolase)—reduced the growth rates of Lewis lung carcinoma (LLC)-derived tumors in ω-3 PUFAs-fed mice and inhibited their metastasis to lung and liver, but a ω-6 PUFA-rich diet produced the opposite effects. These findings, therefore, reveal a novel mechanism whereby the oxylipins, particularly epoxides generated from ω-3 PUFAs by the TCDD/AHR/CYP1s/sEH pathway protect against tumor growth and metastasis, while ω-6 oxylipins induce the opposite result, through affecting vascularity, cell proliferation and/or apoptosis [20]. Thus, the adequate balance of PUFAs in the diet is essential, as they are biologically active molecules that play key roles in metabolism, inflammation, cell signaling, and regulation of gene expression in cancer progression. However, until now, there has been no evidence that a diet rich in ω-6 can contribute to a change in the tumor aggressiveness phenotype. Thus, in this work, we used an experimental model of pulmonary squamous cell carcinoma to study the aggressiveness of the tumor in mice fed with diets rich in ω-6 and its relationship with oxylipins.

## 2. Results

### 2.1. ω-6 Feeding Induces Higher Lung Cancer Aggressiveness, De-Differentiation and Increase in Proliferation

The general outline of the experiment is shown in Figure 1. The tumor derived from mice fed with a rich-ω-6 diet presented wide hemorrhage and extensive necrosis areas as compared with control or PUFA-balanced diet-fed mice. The evaluation of the H&E staining sections of LLC-derived tumors from mice fed with the control diet showed mild differentiated epidermal carcinoma, constituted by mantles of polyhedral cells with frequent mitotic figures. Giant cells have extensive eosinophilic cytoplasm, with a single multilobed hyperchromatic nucleus, which is very oddly shaped and strongly nucleated. Tumors from mice fed with the PUFA-balanced diet showed similar histopathological appearance as tumors from mice fed with control diet, except that the tumors had frequent neoplastic apoptotic cells and focal areas of necrosis (Figure 2A). In contrast, mice fed with a ω-6-rich diet developed poorly differentiated tumors, comprised of mantles of spindle cells with sarcomatoid pattern. Many of these cells exhibited severe dysplastic changes, such as big and vesicular nuclei with chromatin clumps and large, acidophilic, and prominent nucleolus, cells were arranged in bundles, with a storiform appearance, showed numerous mitotic figures, many of them atypical, and large areas of necrosis with acute inflammation. All these histological features are indicative of tumor dedifferentiation and higher aggressiveness. 

It has been shown that the positivity of the Pan-cytokeratins is a confirmation of the carcinomatous character of the tumor together with predominant spindle or giant cells, including the expression of cytokeratins 5 and 6 (CK5/6) [21,22,23]. Therefore, we measured the expression of these cytokeratins. We show representative micrographs of CK5/6 immunostaining (Figure 2B). We found a clear overexpression of those cytokeratins in the group of mice fed with a ω-6-rich diet as compared with mice fed with normal chow or with PUFA-balanced diet, where we observed only weak staining. The increase in CK5/6 expression was statistically significant, * *p* < 0.05. These findings confirm the epithelial nature of this sarcomatoid carcinoma, and its squamous dedifferentiation associated with high intake of ω-6 PUFAs. 

Since the areas of necrosis are related to high proliferative neoplastic activity, which is a hallmark of highly malignant tumors, we evaluated cellular proliferation through mitotic index measurement in the H&E staining slides and the expression of mini-chromosome maintenance protein 2 (MCM2) by IHC. Figure 2C shows that mice fed with normal chow show a 2.4% mitotic index, very similar to that seen in mice fed with balanced diet tumor (2.6%). In contrast, tumors from mice fed with a ω-6-rich diet exhibited a higher percent of mitotic index (6.6%) (** *p* < 0.01). To validate these results, we performed MCM2 detection by immunostaining (Figure 2D). As expected, mice fed with a ω-6-rich diet showed higher expression of MCM2 than mice fed with control or PUFA-balanced diets (** *p* < 0.01 and * *p* < 0.05, respectively).

### 2.2. High Intake of ω-6 PUFA Inhibits Caspase-Dependent Apoptosis

Since the histological evaluation demonstrated large areas of necrosis, we analyzed whether the high levels of ω-6 PUFA could activate the apoptotic process and contribute to areas of cell death. Therefore, the presence of active caspase-3, -8 and -9 was evaluated in tumor tissue by IHC in the three different mice groups. In Figure 3, we present a representative micrograph of tumors derived from a ω-6 rich diet, PUFA-balanced diet, and control diet. The results demonstrated a significant increase in active caspases 3 (Figure 3A) (* *p* < 0.05), 8 (Figure 3B) (* *p* < 0.05) and 9 (Figure 3C) (* *p* < 0.05) positive cells in the mice group fed with the PUFA-balanced diet as compared to those who received a control diet or ω-6 rich diet. 

### 2.3. ω-6-Rich Diet Increases Angiogenesis of Tumor Cells

Growth progression of malignant solid tumors depends on vascularization and angiogenesis, with vascular endothelial growth factor A (VEGF-A) being a significant contributor [24]. Thus, we evaluated VEGF-A expression in mice fed with the different diets. Figure 4A shows representative micrographs of VEGF-A immunostaining in LLC-derived tumor. There is a significantly higher expression of this factor in mice fed with a ω-6-rich diet, mostly in the cytoplasm, than in mice fed with control or PUFA balanced diet (* *p* < 0.05 and ** *p* < 0.01, respectively). In addition, neo-vascularization was determined by the expression of CD31 (platelet endothelial cell adhesion molecule). In comparison with mice fed with a control or balanced diet, the number of CD31+ immunostained vessels were significant higher in mice fed with the ω-6-rich diet (** *p* < 0.01 and * *p* < 0.05, respectively) (Figure 4B). 

### 2.4. ω-6 Rich Diet Induces Oxylipin Generation Associated with Tumor Aggressiveness

We analyzed the profile of oxylipins involved in the change in tumor aggressiveness phenotype. Since the greatest difference was observed between the groups of animals that received a balanced diet and ω-6 rich diet, we evaluated the levels of 66 oxylipins through LC/MS/MS in the plasma of five animals fed for four weeks with those diets without tumor induction to evaluate the basal level of oxylipins that the diets generate in the animals and that may affect tumor development. We compared the oxylipins levels between both diets through the Mann–Whitney U test and selected those with *p* < 0.05. Data are presented as the mean ± SEM and levels of significance are indicated as follows: * *p* < 0.05, ** *p* < 0.01, *** *p* < 0.001 (Table 1 and Table 2), the levels of all oxylipins evaluated (66) are shown in Appendix A. Oxylipins were classified according to their parental PUFAs and enzymatic pathway origins. Interestingly, mice fed with a ω-6-rich diets showed higher levels of oxylipins derived from COX and LOX pathways compared to CYP450 oxylipins in plasma. COX-dependent ARA oxylipins that increased included Thromboxane B_2_ (TXB_2_) and prostaglandin D_2_ (PGD_2_). Lipoxygenase products included lipoxygenase A4 (LXA_4_), 12-oxoeicosatetraenoic acid (12-oxo-ETE) and hydroxyeicosatetraenoic acids (HETEs). The cytochrome P450 pathway presented an increase in the diols (DiHETrEs). Likewise, LA showed high levels of LOX oxylipins such as hydroxy octadecaenoic acids (HODEs) (Table 1). In addition, we found an increase in levels of epoxy-octadecenoic acids (EpOMEs) and dihydroxy-octadecadienoic acids (DiHOMEs), although they were not significantly different between each diet group (Appendix A). 

Additionally, mice fed with balanced diet showed higher levels of oxylipins derived from the CYP450 pathway compared to COX and LOX routes in plasma. EPA oxylipins that increased through the COX pathway included prostaglandin D_3_ (PGD_3_). There was an increase in hydroxyeicosapentaenoic acids (HEPEs) levels in the LOX pathway, and the cytochrome P450 pathway showed high levels of epoxides (EpETEs) and diols (DiHETEs). In addition, DHA presented an increase in epoxides (EpDPEs) and diols (DiHDPEs) derived from the CYP450 pathway. All the changes between mice fed with a ω-3-rich diet and PUFAs are presented in Table 2. These data show that the diet generated an oxylipin microenvironment that can influence the aggressiveness of the tumor.

### 2.5. ω-6-Rich Diet Increases the Aggressiveness of Tumor Cells

To support our findings, we subsequently evaluated different markers involved in tumor aggressiveness. The high expression of the transcription factor Yin Yang 1 (YY1) has been shown to positively correlate with large tumors, poor differentiation, and metastasis to lymphoid nodules in patients with lung cancer [25]. in line with this, our results showed significantly higher induction of YY1 protein expression in mice fed with a ω-6-rich diet than in mice fed with the control diet or with a balanced diet (** *p* < 0.01 and * *p* < 0.05, respectively) (Figure 5A). In addition, we evaluated the expression of COX-2 (Figure 5B) and TGF-β (Figure 5C), because they are target genes of YY1, and the expression of both genes was significantly higher in the neoplastic tissue from mice fed with a ω-6-rich diet (* *p* < 0.05). The expression of YY1 was primarily in the nucleus, while the immunostaining of COX-2 and TGF-β were mostly in the cytoplasm. To correlate the IHC findings with gene expression, the YY1, COX-2 and VEGF-A mRNA levels were determined by RT-PCR. Indeed, mice fed with a ω-6-rich diet group showed higher expression of these genes than mice fed with control diet or balanced diet (* *p* < 0.05 and ** *p* < 0.01, respectively) (Figure 5D–F). Together, all these results strongly suggest that the ω-6-rich diet enhanced the malignancy of this tumor phenotype.

To better understand the mechanism through which ω-6 PUFAs increase the expression and activation of YY1 and its target proteins, we evaluated the expression of NF-κB p65. Several reports indicated that ω-6 PUFAs as well as their metabolites activate the NF-κB pathway [26,27], which in turn can transcriptionally activate YY1 [28]. Importantly, we found that the mice that received a diet rich in ω-6 presented an increase in the nuclear expression of NF-κB p65 in tumor cells compared to the control diet and balanced diet groups (* *p* < 0.05) (Figure 6). These findings strongly suggest that the NF-κB pathway is activated when there is a high intake of ω-6 PUFAs, activating its target genes, including YY1, thereby enhancing the aggressive phenotype.

## 3. Discussion

The effects of nutrition in the cancer process are very broad, and it is clear that diet plays an important role in the development of cancer, as shown by the observation that 30–40% of all cancers can be prevented by appropriate diets [29]. An adverse role of high ω-6 PUFAs diet in cancer epidemiology and its outcomes have been extensively evaluated in clinical and animal studies [13]. In this regard, several reports demonstrated that a high intake of ω-6-PUFAs, such as ARA and LA, which are common constituents of the Western diet, is associated with cancer progression [8]. In contrast, ω-3-PUFAs, such as ALA, EPA and DHA are generally correlated with cancer protection [13]. Recently, our group showed that TCDD, a prototypical AHR ligand, plus TPPU (an inhibitor of soluble epoxide hydrolase), affect the growth and metastasis of lung tumors in either a positive or negative way, depending on the concentrations of ω-3 and ω-6 PUFAs in the diet. This study revealed a novel mechanism whereby AHR activation as well as CYP1s enzymes can impact cancer progression through the accumulation of oxylipins, particularly epoxides, which confirms the benefits of a high ω-3 PUFA diet and the risk of a high ω-6 PUFAs diet in promoting tumor growth and metastasis in a lung cancer model [20]. However, the role of a ω-6 rich diet in tumor aggressiveness has not been evaluated

In this study, we demonstrated for the first time that a ω-6-rich diet, as compared with control diet or a PUFA-balanced diet, induces in the LLC-derived tumor a sarcomatoid dedifferentiation. This differentiation is currently thought to represent a transformation to a malignancy of a higher grade, characterized by a spindle cell histologic appearance with ultrastructural and immunohistochemical evidence of epithelial and mesenchymal differentiation. Tumors with this histologic appearance have been referred to as metaplastic carcinoma [30]. That finding is interesting if we consider that lung sarcomatoid cancer is a rare and poorly differentiated histologic type of NSCLC with a component of sarcoma or sarcoma-like differentiation. This histologic variant is associated with significantly worse prognosis [31]. Martin et al. reported that the 5-year survival rate for lung sarcomatoid cancer patients was 24.5% compared with 46.3% for NSCLC patients; in addition, median time to recurrence was 11.3 months for lung sarcomatoid cancer patients and 61.4 months for NSCLC patients [32]. Moreover, metastatic patients with sarcomatoid carcinomas have been reported to respond poorly to conventional chemotherapy [33,34] and exhibit chemoresistance [35]. 

As previously mentioned, TCDD, a prototypical AHR ligand, affects the growth and metastasis of lung tumors in either a positive or negative way, depending on the concentrations of ω-3 and ω-6 PUFAs in the diet [20]. In this model, we did not find differences in the control groups of mice (that did not receive TCDD) in the tumor growth between the mice group that received of ω-3-rich diet (ω-3 to ω-6 ratio of 1:1.1) vs. the group that was fed with a ω-6-rich diet (ω-3 to ω-6 ratio of 1:20). This would appear to be contradictory with the results obtained in the present study. However, in that previous study, we only allowed the tumors to reach a size no more than 2500 cm^3^ and we did not evaluate histopathological changes in mice fed with ω-6 rich diet, at this time, since it was not the objective of that study. In contrast, in the present study, the tumors were removed when they reached at least 3000 cm^3^. This suggests that the histopathological changes that denote an aggressive phenotype in the tumors of mice that received a diet rich in ω-6 require a longer time for this phenotype to be clearly manifested.

Furthermore, positive staining for CK5/6 demonstrates the epithelial origin of this sarcomatoid carcinoma [21,22,23]. The results obtained in this study regarding the expression of those cytokeratins supports the fact that it occurs by squamous dedifferentiation and higher aggressiveness caused by high intake of ω-6 PUFAs. In addition, we also demonstrated that both mitotic index and MCM2 expression were higher in the mice group that received ω-6-rich diet. Those parameters are extensively used as neoplastic proliferation markers. [36,37,38,39]. Thus, neoplastic cells dedifferentiation and high index of cellular proliferation of tumors from mice fed with the ω-6-rich diet support the change to the more aggressiveness phenotype. 

It has been reported that the inhibition of apoptosis leads to the promotion of cancer [40]. Several reports demonstrated the role of PUFAs in the apoptotic process, showing evidence that attributes a pro-apoptotic role to type ω-3 PUFAs. However, ω-6 PUFAs demonstrate a contradictory effect as some authors indicate their role in induction of cell death while others state otherwise [41,42,43]. In this study, we demonstrated that ω-3 and ω-6 PUFAs are involved in the apoptosis mechanisms in our lung cancer model. Interestingly, we found that the consumption of a balanced diet in PUFAs can activate the caspase-3, -8 and -9-dependent apoptotic process. Several studies support the role of ω-3 PUFAs in the activation of the extrinsic pathway of apoptosis in promyelocytic leukemia cells (HL-60) [44], as well as in ER+ breast cancer cells [45], where both studies showed an increase in caspase-8. Other studies also support the activation of the intrinsic pathway [43]. Even Ryadi et al. demonstrated a “crosstalk” between both pathways mediated by treatment with C20E (an analog of EPA) in human MDA-MB-231 breast cancer cells. C20E through TNFR1 causes the activation of the ASK1-MKK4/JNK/p38MAPK signaling pathway, promoting the breakdown of Bid (tBid), leading to MOMP and the activation of the mitochondrial pathway [46,47]. On the other hand, it was very surprising for us to find very low level of expression of active caspases 3, 8 and 9, despite the high concentration of ω-6 PUFAs (ω-6/ω-3 20:1) that this diet contains, despite the reports made by other authors where they indicate an apoptosis-inducing effect both by the intrinsic [48,49,50] and extrinsic [51] pathways of the ω-6 PUFAs (ARA, LA and GLA). This minimal activation of the apoptotic pathway may explain why the mice in this group show greater aggressiveness. Additionally, our results agree with studies that support the inhibitory role of ARA [52,53], as well as its derived metabolites (PGE2, PGE4 12-HETE, 14-HETE, etc) [54] in apoptosis. It is worth mentioning that ARA and LA can also activate survival and proliferation pathways [55], which may be preventing the activation of apoptosis in this murine model. Moreover, our results showed that LLC tumors from mice fed with the ω-6-rich diet presented high levels of VEGF-A and CD31, contributing to the aggressiveness of the tumor. However, more experiments need to be carried out to further investigate the detailed pathways.

Chronic inflammation is one of the most important factors leading to carcinogenesis. Obesity, inflammatory bowel disease (IBD), chronic colitis and pancreatitis are some specific examples of inflammatory conditions that are known risk factors for cancer with shared mechanisms involving altered levels of oxylipins [56,57]. Oxylipins are PUFA oxidation products formed via one or more mono- or dioxygen-dependent reactions, and they are major mediators of PUFA effects in the body. In this study, we evaluated, through LC/MS/MS analysis, the oxylipins levels in plasma, to identify which oxylipins are involved in the aggressivity phenotype after high ω-6-PUFAs consumption. Interestingly, in mice fed a ω-6-rich diet, we found a significant increase in several ARA oxylipins, compared with the mice fed with a balanced diet, especially the ones where the COX pathway is involved (TXB2 and PGD2). In addition, mice fed with the PUFA-balanced diet presented an increase in PGD3. Several reports demonstrated that the 2-series of PGs, principally PGE2, act as potent mediators of inflammation and cell proliferation [58], as well as angiogenesis and invasion [59], which further stimulate tumor cell growth and progression of NSCLC [60]. In contrast, PGE3 inhibited the proliferation of the A549 lung adenocarcinoma lung cell line [61]. In addition, mice fed the ω-6-rich diet showed high levels of the ARA oxylipins derived from LOX pathway, including LXA4, 12-oxo-ETE, HETEs and DiHETrEs. In particular, it has been reported that HETEs participate in the angiogenesis, proliferation, and metastasis processes [62]. For instance, Liu et al. found high 15-HETE levels in NSCLC patients compared with controls, and they propose it as a possible marker [14]. Moreover, mice fed the PUFA-balanced diet presented high HEPEs levels; Vang et al. showed that 15-HEPE inhibits cellular growth and ARA metabolism in human prostatic adenocarcinoma cells [63]. These studies support our finding that there was an increase in proliferation and angiogenesis in mice fed the ω-6-rich diet compared to the PUFA-balanced diet and the control mice groups. In addition, in mice fed the ω-6-rich diet, we found elevated levels of LA oxylipins such as EpOMEs and DiHOMEs, although they were not significant, it is important to note that these oxylipins participate in inflammatory processes, vascular permeability, neutrophil chemotaxis, as well cytotoxic effects in advancing acute and chronic inflammatory diseases [64]. Interestingly, McReynolds et al. reported that patients with severe COVID-19 who show evidence of hyperinflammation mainly showed high levels of 9,10 and 12,13-DiHOMEs, compared with healthy controls [65]. Moreover, mice fed with the PUFA-balanced diet showed higher levels of oxylipins derived from the CYP450 pathway, and we found an increase in EPA epoxides (EpETEs). Several lines of evidence have shown the participation of 17,18-EpETE in the decrease in cell proliferation in brain endothelial cells through inhibition of the Cyclin D1/p38/MAPK axis [66]. Additionally, this enhances caspase-3 activity and activates JNK signaling, leading to cyclin D1 downregulation and cell cycle arrest in G1-phase in MDA-MB-231 breast cancer cells [46]. Furthermore, we found high DHA epoxide (EpDPEs) levels in our model mice. Some studies showed that 19,20-EpDPE decreased the growth in a breast cancer murine model [67] and decreased angiogenesis by inhibiting the expression of VEGF-A and Fibroblast Growth Factor 2 (FGF2) in umbilical cord endothelial cells [16]. In addition, 16,17-EpDPE and 19,20-EpDPE plus t-AUCB (inhibitor of soluble epoxide hydrolase) decrease LLC metastasis in a murine model [16,68,69] and Xia et al. showed that ω-3 epoxy acids reduce microvessel density (MVD) (CD34 expression) in pancreatic carcinoma [70]. These studies support our results and could explain in part why the mice fed with the ω-6-rich diet have a more aggressive phenotype.

To support our results regarding the change in the aggressivity phenotype after rich ω-6-PUFA diet intake, we analyzed several markers involved in aggressiveness. Interestingly, in comparison with mice fed with control diet or balanced diet, tumors from animals fed with the ω-6-rich diet exhibited elevated expression of the transcription factor YY1. This observation is consistent with past findings that considering that YY1 overexpression is linked with many cancers and its levels largely correlated with cancer progression, metastasis, drug resistance and poor prognosis [71]. More specifically, YY1 can transcriptionally regulate COX-2 [72] and overexpression of the COX-2 gene is linked to the pathogenesis of various types of cancer, including lung cancer [73]. Interestingly, we also detected in the tumors from mice fed with the ω-6-rich diet higher expression of COX-2, which also could explain in part the increase in COX-2-dependent oxylipins in this mice group. In addition, COX-2 contributes to the regulation of angiogenesis by various genes, including VEGF-A [74], as demonstrated by Marrogi et al. in patients by the positive correlation between COX-2 and VEGF-A protein expression levels and MVD. The clinical stage of both squamous and non-squamous carcinomas was associated with MVD levels [75]. Additionally, Wu et al. reported that overexpression of both COX-2 and mPGES-1 adversely affects postoperative overall survival and disease-free survival in NSCLC [76] and Giaginis et al. showed that enhanced COX-2 protein expression was significantly associated with the presence of lymphovascular invasion and increased tumor proliferative capacity in NSCLC patients [77]. Indeed, our results on the expression of COX-2 and VEGF-A angiogenesis in mice fed the ω-6-rich diet compared to high ω-3 and control mice groups are consistent with that observation. Moreover, YY1 was demonstrated to induce the EMT phenotype through TGF-β in lung tumor cells [78]. Furthermore, TGF-β can regulates COX-2 [79]. TGF-β, in healthy cells and early-stage cancer cells, has tumor-suppressor functions, including cell-cycle arrest and apoptosis, but it can promote metastasis and chemoresistance in late-stage cancer [80]. Our results demonstrated that the tumor from mice fed with the ω-6-rich diet, compared with control or balanced diet, also overexpressed TGF-β. TGF-β overexpression could be part of a feedback system in our model that supports the aggressiveness phenotype in mice fed with a ω-6-rich diet. For the first time, our results demonstrated that a high intake of ω-6 PUFAs increases the expression and activation of YY1, enhancing the aggressive phenotype in this group of animals. A possible mechanism by which these lipid mediators could activate the expression of YY1 would be through an indirect pathway mediated by NF-κB, which transcriptionally regulates YY1. Several studies reported that PGE2, when binding to its receptor EP2, can activate the NF-κB pathway in an ovarian cancer cell model (SKOV3 and OVCAR3) [81] and in a mouse model of intracranial aneurysm [27]. In the same way, we showed for the first time that the high consumption of ω-6 PUFAs causes the translocation of NF-κB p65 to the nucleus, activating its target genes, including YY1, enhancing the aggressiveness phenotype. This evidence supports our findings and strongly suggests that a high intake of ω-6 PUFAs leads to a more aggressive phenotype of lung cancer mediated through pro-tumor oxylipins and high expression of VEGF-A, CD31, YY1, COX-2 and TGF-β.

## 4. Materials and Methods

### 4.1. Cell Line

The LLC cell line was purchased from the American Type Culture Collection (ATCC) (Manassas, VA, USA). This mouse cell line is a highly malignant type of epidermoid lung carcinoma and becomes extremely hemorrhagic [82]. LLC cells were cultured in alpha Minimal Essential Medium (α-MEM) (Gibco, Waltham, MA, USA), supplemented with 10% fetal bovine serum (FBS) (Gibco), and Antibiotic and Fungizone (Gibco). Cells for in vivo injections had greater than 90% viability and were harvested from sub-confluent cultures using 0.05% trypsin-EDTA (Gibco), neutralized with culture medium at 4 °C and washed 3 times in serum-free α-MEM before re-suspension in serum-free α-MEM. 

### 4.2. Experimental Model 

C57BL/6 wild-type mice aged 5 weeks old were purchased from the Jackson Laboratory (Bar Harbor, ME, USA). Mice were housed up to five per cage in independently ventilated cages at a constant temperature (20~22 °C) and a specific pathogen-free environment, in the animal facility at University of California Los Angeles (UCLA) and Hospital Infantil de Mexico Federico Gomez (HIMFG). Mice were divided randomly into three different groups with 10 mice each, according to the diet: animals fed with control diet (PURINA, 5001), ω-6-rich diet with ω-6/ω-3 ratio of 20:1 (Envigo Teklad, TD. 130321 [20]) (this diet more closely resembles the ω-6/ω-3 ratio in the typical Western diet) and a PUFA-balanced diet with ω-6/ω-3 ratio of 1.1:1 (Envigo Teklad, TD. 140429 [20]). (This ratio is optimal for the human health [83].) Fatty acid composition of diets is shown in Table 3. Importantly, mice maintained their weights equally on all diets (data not shown). Mice consumed equal amounts of calories per day on each diet. The food was changed twice per week. They were fed ad libitum with these diets during 4 weeks prior to cell injection. Five mice were sacrificed after 4 weeks, and the plasma was obtained for subsequent measurement of oxylipins through LC/MS/MS. The remaining five mice were subcutaneously injected in the right flank with a single-cell suspension of 2 × 10^5^ LLC cells. Two weeks after cell injection, tumor volume was assessed every day with an electronic caliper until they reached the volume of approximately 3000 mm^3^ when mice were euthanized. Tumor volumes were determined using the formula: volume = (Lxw^2^)/2) × (¾π) [84]. 

Tumors were divided in two halves; one half was fixed by immersion in formalin buffered saline (Thermo Fisher Scientific. Waltham, MA, USA) and embedded in paraffin for histological and immunohistochemical analysis. The other tumor half was cut in small pieces and immediately put into liquid nitrogen for subsequent measurement of mRNA through RT-PCR. The model is shown in Figure 1. The experiments were approved by the UCLA Institutional Animal Care and Use Committee (IACUC) and HIMFG, and the animals were cared for in accordance with institutional guidelines. 

### 4.3. Hematoxylin and Eosin Staining

Sections of 4 µm of tissue were dewaxed in xylol (J.T. Baker, Radnor, PA, USA) and rehydrated through a series of alcohols. Subsequently, sections were immersed in hematoxylin for one minute, followed by three washes in ethanol-HCl, one wash in water, one wash in 70% ethanol (J.T. Baker), immersed in eosin for two minutes, and then the tissue was dehydrated through a series of alcohols. Finally, the slides were covered with resin and the coverslip allowed them to dry at room temperature. An expert pathologist selected representative areas from each sample for observation. 

### 4.4. Mitotic Index 

From each hematoxylin and eosin (H&E) staining tumor tissue, 10 random fields were taken at magnification of 40× and the number of mitoses was counted. The average was obtained and reported as a mitotic index through the following formula: Mitotic index = Number of mitosis/10. 

### 4.5. Immunohistochemical Analysis and Digital Pathology 

The sections of tissue measuring 4 µm were dewaxed in xylol (J.T. Baker) and rehydrated through a series of alcohols (xylene, 100% ethanol, 90% ethanol, 70% ethanol and distilled water). Antigen retrieval was performed. Endogenous peroxidase activity was blocked in 10% hydrogen peroxide solution (J.T. Baker), and slides were incubated with specific primary antibodies (Table 4). Finally, sections were incubated with the ImmPRESS HRP Horse Anti-Rabbit IgG Polymer Detection Kit Peroxidase from Vector Laboratories (Burlingame, CA, USA) and DAB detection system (Vector Laboratories), counterstained with hematoxylin, dehydrated, and mounted.

As previously described [6], the ImmunoHistoChemistry (IHC)-stained sections were digitized using an Aperio ScanScope CS (Leica BioSystems, Nussloch, Germany) which obtains 20× digital images with high resolution (0.45 um/pixel). Images were viewed using an ImageScope (Aperio, San Diego, CA, USA) to analyze and quantify marker expression. An algorithm was developed for each tissue to quantify total and the nuclear protein expression. The ImageScope allowed the setting of a threshold for color saturation as well as upper and lower limits for intensities of weak, moderately, and strong positive pixels. To ensure that the settings were appropriate for analysis of different tissues, randomly selected cores on the parameters were customized to differentiate between negative (blue), low (yellow), moderate (orange), and strongly (red) stained. The raw data included the number of positive pixels and the intensity of positive pixels, which were normalized to number of total pixels counted in µm^2^. Data are presented as total density/µm^2^ analyzed in a total area of 10,000 µm^2^. 

### 4.6. Oxylipin Profiling

Briefly, 200 uL of mice plasma was spiked with 9 isotope labeled internal standard before extraction. Solid phase extraction (SPE) was used to clean up the samples and the samples were eluted out from Oasis HLB SPE (Waters, Milford, MA, USA) cartridge by 0.5 mL of methanol and 1 mL of ethyl acetate. The eluants were then concentrated in a speedvac and reconstituted to 50 µL of solution for the followed LC/MS/MS measurements. 

The LC/MS/MS measurements were made using a Sciex 6500+ QTRAP system (Sciex, Redwood, CA, USA) hyphenated to a Waters Acquity LC system. The mass spectrometer was operated under the scheduled MRM mode with electrospray ion source. All the parameters were optimized using the authentic standards and the quantification was carried out against the calibration curve from 0.25 to 800 nM solutions [85,86,87]. 

### 4.7. RT-PCR Analysis

From each frozen tumor tissue, the RNA was extracted using the Trizol and chloroform. Then, 1 µg of RNA was used for cDNA synthesis in a volume of 20 µL. For gene evaluation, we used real time polymerase chain reaction (RT-PCR) with the Universal Taqman Master Mix II kit from Applied biosystems (Waltham, MA, USA). All the TaqMan probes were purchased from Applied Biosystems (Table 5). The relative value of expression of the genes was calculated using the ∆Ct method comparing the expression of mRNA of COX-2, VEGF-A and YY1 related to GAPDH as an endogenous control. 

### 4.8. Statistical Analysis 

Data are presented as the mean ± SD or SEM and analyzed using a Mann–Whitney U test for comparison between two independent groups with each other. Kruskal–Wallis analysis was used for more than two independent groups using the GraphPad Prism Version 5.0 software (San Diego, CA, USA). Levels of significance are indicated as follows: * *p* < 0.05, ** *p* < 0.01, *** *p* < 0.001. 

## 5. Conclusions

In the present study, we demonstrate for the first time that a high intake of ω-6 in the diet enhances the malignancy of tumor cells when compared with animals fed with a balanced diet (1.1:1 ω-6/ω-3 PUFAs) or with the control diet. Various cellular biological activities and molecular targets exemplified this increase in tumor aggressiveness, such as histological changes on tumor dedifferentiation, increase in cell proliferation, enhanced angiogenesis, and increased levels of oxylipins, including PGs, HETEs and HODEs that mediate pro-tumor processes. In addition, the ω-6 rich diet mediated the nuclear expression of NF-κB p65 that correlates with an increase in the expression and activation of YY1 that promotes the transcription of COX-2 and TGF-β, finally promoting a phenotype of greater tumor aggressiveness. A PUFA-balanced diet induces apoptosis-dependent activation of caspase-3, -8 and -9, which could partially explain why this group of mice shows moderate aggressiveness. The contribution of all these components could be part of a complex feedback system in our model that supports the aggressiveness phenotype in mice fed with the ω-6-rich diet (Figure 7). 

These findings can contribute to proposed novel therapeutic modalities based on PUFAs diet, which are likely to improve the treatment and clinical management of aggressive lung cancer, supporting the importance of intake of a balanced ratio of ω-3/ ω-6 PUFAs.

## Figures and Tables

**Figure 1 ijms-23-06179-f001:**
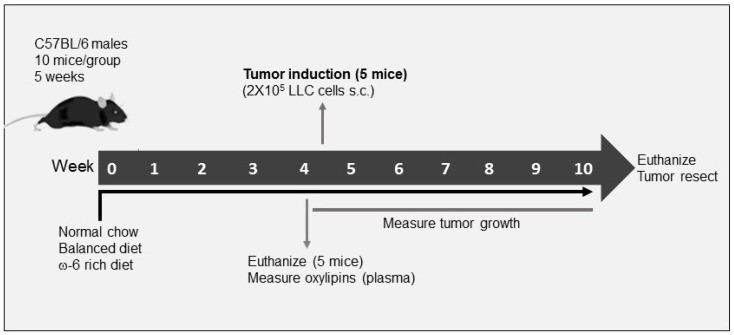
Murine model of the effect of dietary PUFAs of type ω-6 and ω-3 on tumor aggressiveness. Male C57BL/6 mice received a control, balanced or ω-6 rich diet throughout the experiment. After four weeks, five of the mice from each group were sacrificed and plasma was obtained for further analysis of oxylipins by LC/MS/MS. The remaining five animals were administered subcutaneously (sc) with 2 × 10^5^ Lewis lung carcinoma (LLC) cells. The volume of the tumor was evaluated and when it reached a size at least 3000 mm^3^, the mice were sacrificed. The tumor was then resected and divided into two parts: one for histology analyses and the other for RT-PCR.

**Figure 2 ijms-23-06179-f002:**
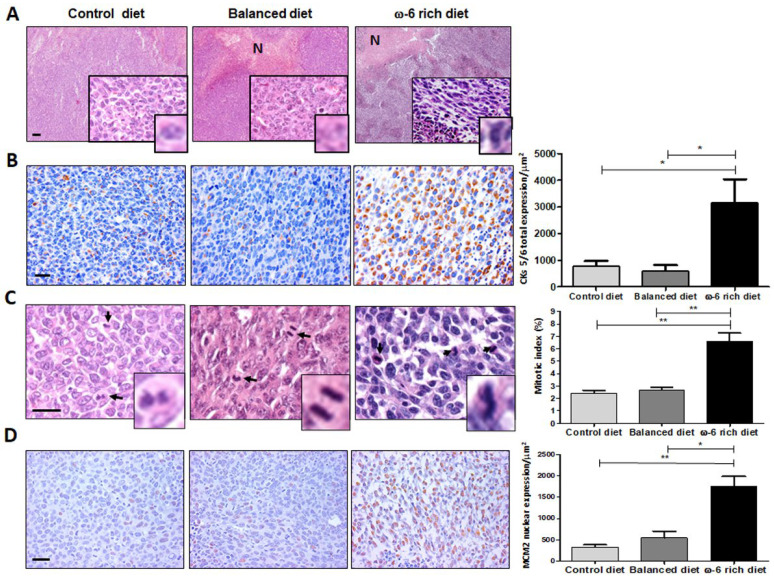
ω-6 PUFAs induce a phenotype of greater tumor aggressiveness and increased cell proliferation. (**A**) Representative photomicrographs of cross sections of tumor dyed with H&E from the different groups of mice fed control diet, PUFA-balanced diet, and ω -6 rich diet. (**B**) Immunostaining and analysis of CK5/6 expression (* *p* < 0.05). (**C**) Mitotic index analysis, statistical differences are shown (** *p* < 0.01). (**D**) Immunostaining and analysis of MCM2 expression, statistical differences are shown (** *p* < 0.01 and * *p* < 0.05). Identifiers: (N) necrosis areas, (black arrows) cells undergoing a mitotic process. (**A**) Scale bar: 200 μm. (**B**–**D**) Scale bar: 20 μm. The data are expressed as the mean + SEM, *n* = 5.

**Figure 3 ijms-23-06179-f003:**
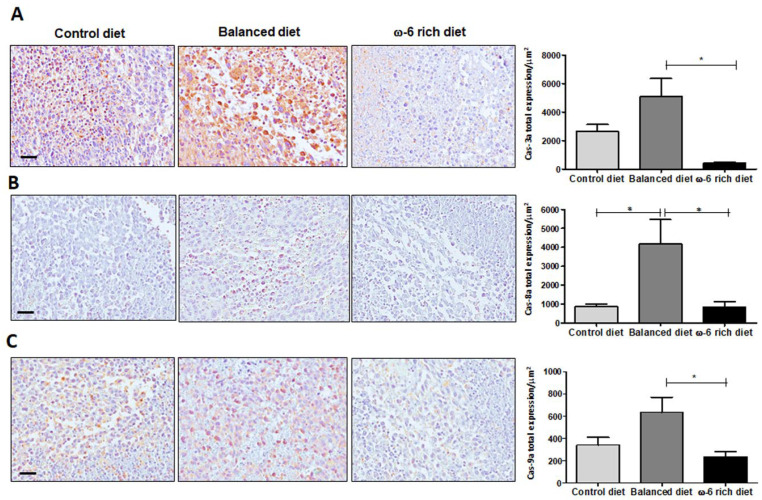
High consumption of PUFA ω-6 inhibits caspase-3, -8 and -9-dependent apoptosis. (**A**) Representative photomicrographs and analysis of active caspase-3 expression analyzed by immunohistochemistry, (* *p* < 0.05). (**B**) Active caspase-8 immunostaining, (* *p* < 0.05). (**C**) Active caspase-9 immunostaining, statistical differences are shown (* *p* < 0.05). Scale bar: 20 µm. Data are expressed as the mean + SEM, *n* = 5.

**Figure 4 ijms-23-06179-f004:**
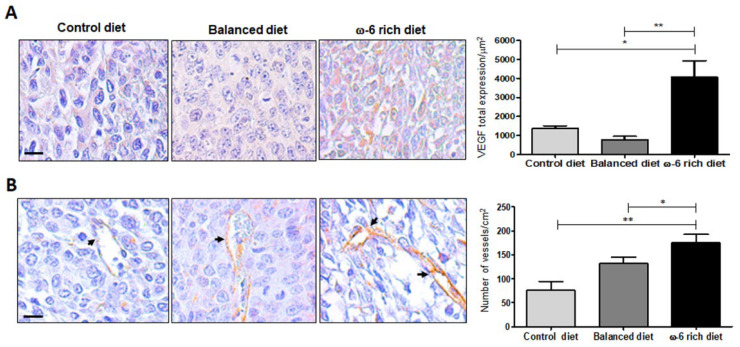
ω-6 PUFAs induce increased vascularization in tumors. (**A**) Representative example of tumor sections and analysis of VEGF-A expression, evaluated by immunohistochemistry, statistical differences are shown (* *p* < 0.05 and ** *p* < 0.01). (**B**) Quantification of blood vessels immunostained with CD31, in the different groups of mice fed with control diet, balanced diet and diet rich in ω-6, on the left side are representative micrographs of the blood vessels denoted by black arrows, right side, the analysis of blood vessels per cm^2^ is shown, (* *p* < 0.05 and ** *p* < 0.01). Scale bar: 10 µm. Values represent mean + SEM, *n* = 5.

**Figure 5 ijms-23-06179-f005:**
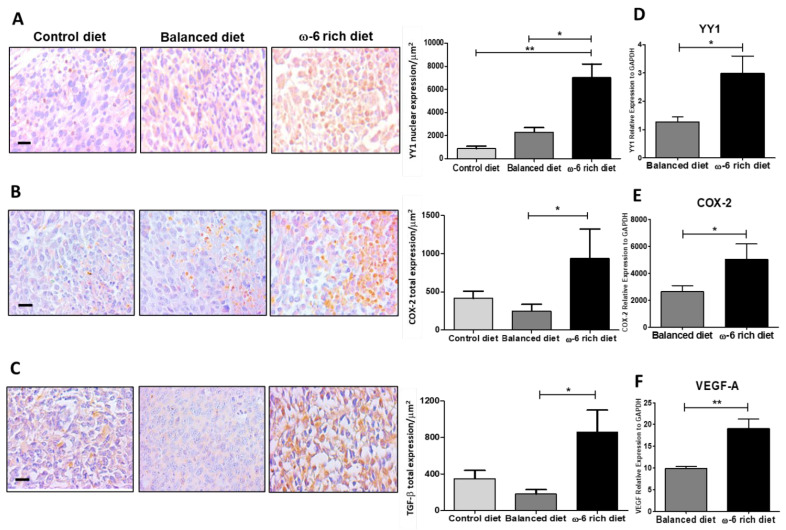
High intake of ω-6 PUFA induces the expression of aggressiveness markers. (**A**) Representative photomicrographs and analysis of YY1 expression (* *p* < 0.05 and ** *p* < 0.01). (**B**) COX-2 immunostaining, * *p* < 0.05. (**C**) TGF-β immunostaining (* *p* < 0.05) in the groups of mice fed with control diet, balanced diet, and diet rich in ω-6. Representative microphotographs of the expression of the different markers are shown on the left. On the right side, the quantification of the stains can be seen, the data represent the mean + SEM, *n* = 5. Scale bar: 20 µm. (**D**) Analysis of YY1, (**E**) COX-2, and (**F**) VEGF-A mRNA in the groups of mice fed a balanced diet and a diet rich in ω-6. Data represent mean + SEM, *n* = 5, * *p* < 0.05, ** *p* < 0.001. Results of three independent experiments are shown.

**Figure 6 ijms-23-06179-f006:**
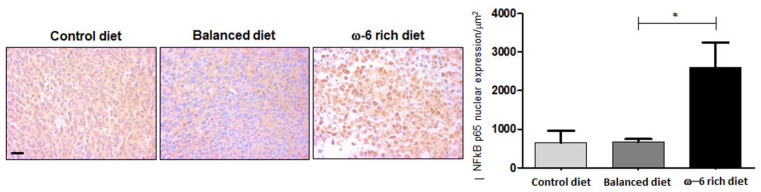
High intake of ω-6 PUFA induces nuclear expression of NF-κB p65. Analysis of p65 expression. Groups: control diet, balanced diet and diet rich in ω-6. Representative photomicrographs of NFκB p65 expression are shown on the left. On the right side, the quantification of the stains can be seen; the data are expressed as the mean + SEM, *n* = 5, * *p* < 0.05. Scale bar: 20 µm.

**Figure 7 ijms-23-06179-f007:**
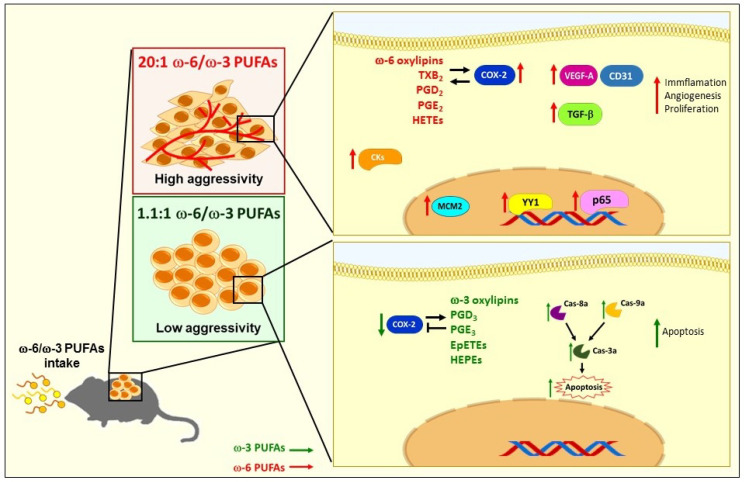
Model of the effect of ω-6 and ω-3 PUFAs on tumor aggressiveness in a murine model of lung cancer. Our study proposes that high intake of ω-6 in the diet increases the aggressiveness of tumor cells compared to animals fed a balanced diet or control diet. This was reflected in histological changes, in tumor dedifferentiation and increased cell proliferation and angiogenesis. It additionally increases the nuclear expression of NFκB p65 that correlates with an increase in the expression and activation of YY1 that promotes the transcription of COX-2 and TGF-β. On the other hand, a balanced diet induces apoptosis dependent on caspase-3, -8 and -9, which could partly explain why this group of mice shows moderate aggressiveness.

**Table 1 ijms-23-06179-t001:** ω-6 oxylipins levels in plasma of mice with balanced and ω-6 rich diet.

ω-6 PUFAs	Enzyme	Oxilypins	Balanced Diet (μmol/L)	ω-6 Rich Diet (μmol/L)	*p* Value
**LA**	**COX**	---	---	---	---
	**LOX**	9-HODE	2654.3 ± 174.1	10,285 ± 612.4	***
		13-HODE	2097.4 ± 112.5	8985.5 ± 473.2	***
	**CYP450**	---	---	---	---
**ARA**	**COX**	TXB_2_	11.0 ± 3.5	28.2 ± 9.4	*
		PGD_2_	0.2 ± 0.06	0.5 ± 0.1	*
	**LOX**	LXA_4_	3.0 ± 1.2	7.9 ± 2.5	*
		12-oxo-ETE	142.5 ± 60.4	534.6 ± 177.1	*
		9-HETE	0.01 ± 0.001	1.2 ± 0.4	***
		12-HETE	325.1 ± 201.1	1751.8 ± 400.1	**
	**CYP450**	5,6-DiHETrE	0.6 ± 0.1	1.2 ± 0.1	**
		8,9-DiHETrE	0.7 ± 0.2	1.6 ± 0.1	*
		14,15-DiHETrE	0.8 ± 0.2	1.5 ± 0.2	*

Data are presented as the mean ± SEM, (*n* = 5). Mann–Whitney U test. * *p* < 0.05, ** *p* < 0.01, *** *p* < 0.001.

**Table 2 ijms-23-06179-t002:** ω-3 oxylipins levels in plasma of mice with balanced and ω-6 rich diet.

ω-3 PUFAs	Enzyme	Oxilypins	Balanced Diet (μmol/L)	ω-6 Rich Diet (μmol/L)	*p* Value
**ALA**	**COX**	---	---	---	---
	**LOX**	13-HOTrE	7.6 ± 1.6	0.2 ± 0.03	**
	**CYP450**	---	---	---	---
**EPA**	**COX**	PGD_3_	0.3 ± 0.1	0.08 ± 0.02	*
	**LOX**	5-HEPE	42.3 ± 19.5	4.1 ± 1.4	*
		8-HEPE	33.4 ± 12.4	1.5 ± 0.7	*
		15-HEPE	73.4 ± 33.2	10.0 ± 2.7	*
	**CYP450**	8,9-EpETE	131.7 ± 34.4	5.2 ± 1.7	**
		11,12-EpETE	106.7 ± 27.1	6.0 ± 1.2	**
		14,15-EpETE	184.4 ± 44.0	10.2 ± 1.7	**
		17,18-EpETE	226.2 ± 55.8	14.4 ± 3.0	***
		11,12-DiHETE	0.6 ± 0.05	0.09 ± 0.02	***
		14,15-DiHETE	0.8 ± 0.1	0.1 ± 0.02	**
		17,18-DiHETE	3.9 ± 0.7	1.4 ± 0.2	**
**DHA**	**COX**	---	---	---	---
	**LOX**	---	---	----	---
	**CYP450**	7,8-EpDPE	5353.7 ± 1520.7	1207.2 ± 319.2	*
		10,11-EpDPE	398.5 ± 117.1	91.0 ± 22.6	*
		13,14-EpDPE	256.5 ± 74.3	58.2 ± 14.5	*
		16,17-EpDPE	253.3 ± 79.4	54.4 ± 13.7	*
		19,20-EpDPE	325.1 ± 102.2	61.0 ± 17.2	*
		7,8-DiHDPE	4550.1 ± 771.2	463.9 ± 165.8	**
		10,11-DiHDPE	1.2 ± 0.2	0.4 ± 0.1	*
		13,14-DiHDPE	1.0 ± 0.2	0.4 ± 0.02	*
		16,17-DiHDPE	2.0 ± 0.4	0.7 ± 0.09	*
		19,20-DiHDPE	17.7 ± 7.1	4.9 ± 0.4	*

Data are presented as the mean ± SEM, (*n* = 5). Mann–Whitney U test. * *p* < 0.05, ** *p* < 0.01, *** *p* < 0.001.

**Table 3 ijms-23-06179-t003:** Fatty acid composition of diets.

	Control Diet	Balanced Diet	ω-6 Rich Diet
Saturated Fatty Acids	15.6	50.2	48.2
Monounsaturated Fatty Acids	16.0	8.2	8.9
Polyunsaturated Fatty Acids	15.2	10.5	12.8
**Total of ω-3 PUFAs**	1.9 *	4.1	0.5
Alpha-Linolenic acid (ALA)	---	0.3	0.2
Eicosapentaenoic acid (EPA)	---	2.3	0.2
Docosahexaenoic acid (DHA)	---	1.5	0.1
**Total of ω-6 PUFAs**	12.3	5	12.2
Linoleic acid (LA)	12.2	4.7	11.9
Arachidonic acid (ARA)	0.1	0.3	0.3
**ω-6/ω-3 ratio**	6.4:1	1.1:1	20:1

Fatty Acids Information (g/kg). * The data sheet does not specify the type of ω-3 PUFAs.

**Table 4 ijms-23-06179-t004:** Antibodies used for protein detection by IHC.

Antigen	Dilution	Company	Catalog Number
Active Caspase-3	1:125	Abcam (Cambridge, UK)	ab32042
Active Caspase-8	1:125	Novus (Saint Charles, MO, USA)	NB100-56116
Active Caspase-9	1:125	Abgent (San Diego, CA, USA)	AP7974a
CD31	1:1000	Abcam	ab28364
Cytokeratin 5/6	1:800	Diagnostic Biosystems (Pleasanton, CA, USA)	PDM123
COX-2	1:500	Novus	NB100-689
MCM2	1:500	Abcam	ab4461
NFkB p65	1:100	Abcam	ab7970
TGF-β	1:500	Abcam	ab92486
VEGF-A	1:500	Abcam	ab46154
YY1	1:500	Novus	NBP2-20932

**Table 5 ijms-23-06179-t005:** Probes used for mRNA detection by RT-PCR.

Probe	ID	Company
COX-2	Mm03294838_g1	Applied Biosystems (Waltham, MA, USA)
VEGF-A	Mm00437306_m1	Applied Biosystems
YY1	Mm00456392_m1	Applied Biosystems
GAPDH	Mm05724508_g1	Applied Biosystems

## Data Availability

Not applicable.

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
