# Peer review of "Omega-6 Polyunsaturated Fatty Acids Enhance Tumor Aggressiveness in Experimental Lung Cancer Model: Important Role of Oxylipins"

_ijms, 2022, doi:10.3390/ijms23116179_

Round 1
Reviewer 1 Report
Omega-6 polyunsaturated fatty acids enhance tumor aggressiveness in experimental lung cancer model: important role of oxylipins (ijms-1713947)
PUFAs play crucial roles in maintaining cellular homeostasis, and perturbations in dietary intake or PUFAs metabolism could result in cellular dysfunction and contribute to cancer risk and progression. According to the authors several studies have demonstrated that ω-6 PUFAs induce progression in certain types of cancer.
Authors used an experimental model of pulmonary squamous cell carcinoma to study the aggressiveness of the tumor in mice fed with diets rich in ω-6 and its relationship with oxylipins. The mice fed a diet rich in ω-6 showed a marked increase in proliferation, angiogenesis and pro-inflammatory markers and decreased expression of pro-apoptotic proteins in their tumors. Oxylipin profiling revealed an upregulation of various pro-tumoral oxylipins including PGs, HETEs, DiHETrEs and HODEs.
This study is very interesting, timely and vel planned. Also the results are very good describes, well discussed and are in agreement with other authors. Well thought-out conclusions appear.
Minor suggestion:
- All tables must be better described. Table 1 and 2 – authors should explain which significance levels the individual symbols define. What statistical test was used? From many mice were plasma collected for oxylipin profiling (number N) ?
- Oxylipin profiling should be briefly described.
- What are the limitations of study?
Author Response
Reviewer: 1
PUFAs play crucial roles in maintaining cellular homeostasis, and perturbations in dietary intake or PUFAs metabolism could result in cellular dysfunction and contribute to cancer risk and progression. According to the authors several studies have demonstrated that ω-6 PUFAs induce progression in certain types of cancer.
Authors used an experimental model of pulmonary squamous cell carcinoma to study the aggressiveness of the tumor in mice fed with diets rich in ω-6 and its relationship with oxylipins. The mice fed a diet rich in ω-6 showed a marked increase in proliferation, angiogenesis and pro-inflammatory markers and decreased expression of pro-apoptotic proteins in their tumors. Oxylipin profiling revealed an upregulation of various pro-tumoral oxylipins including PGs, HETEs, DiHETrEs and HODEs.
This study is very interesting, timely and vel planned. Also, the results are very good describes, well discussed and are in agreement with other authors. Well thought-out conclusions appear.
Minor comments
We thank the authors for your kind comments.
- All tables must be better described. Table 1 and 2 – authors should explain which significance levels the individual symbols define. What statistical test was used? From many mice were plasma collected for oxylipin profiling (number N)?
Author response a: The changes were added in table 1 and table 2.
- Oxylipin profiling should be briefly described.
Author response a: We described the oxylipin profile as the reviewer suggested (Lines 507-516).
- What are the limitations of study?
Author response a: Thank you for your comments. Firstly, it is difficult to confirm which mechanism or pathway the PUFAs affect tumor progression. However, many studies have demonstrated that both PUFAs and oxylipins can bind to receptors such as PPAR and EP, and trigger their effects (10.1158/0008-5472.CAN-15-0125).
Secondly, we cannot determine the exact amount of food consumed by each mouse. However, regardless of food consumption, the proportion of PUFAs in each diet is conserved. We are aiming to evaluate the effect of oxylipins on tumor aggressiveness, which depends on the metabolism of the animal in addition to food consumption.

Reviewer 2 Report
Dear Authors I repor my comments:
"However, until now there is no evidence that diet rich in ω-6 can contribute to change of the tumor aggressiveness phenotype
I’m not totally agree, so please ameliorate the bibliography research on this point
"Thus, in this work we used an experimental model of pulmonary squamous cell carcinoma to study the aggressiveness of the tumor in mice fed with diets rich in ω-6 97 and its relationship with oxylipins"
Why did you use this kind of " cancer" for your thesis??
The experimental part about the “ experimental model of pulmonary squamous cell carcinoma to study the aggressiveness of the tumor in mice fed with diets rich in ω-6 and its relationship with oxylipins “ was now well described, further only the number of references 24-26 are mentioned
So please add detailed description of the work up of the samples
So, please provide the lipidomic profiles of the tissue samples. Further, what is the FAME profile? this is important in order to evidence the remodelling effect of omega 6 ; please provide a a very nice GC chromatogram picture highlighting each fatty acid recognized. This is important aspect due the connection between the omega 6( like Arachidonic acid) and oxylipin metabolites
Please perfom the specific statistical correlation between the fatty acid profile , the oxylipin and the signals in order to demonstrate and confirm your thesis
Author Response
Answer to the Reviewer Comments Manuscript ID ijms-1713947
Reviewer: 2
- "However, until now there is no evidence that diet rich in ω-6 can contribute to change of the tumor aggressiveness phenotype
I’m not totally agree, so please ameliorate the bibliography research on this point
Author response a: We agree that there are numerous studies that relate the high intake of w-6 PUFAs with pro-tumor processes (proliferation, angiogenesis, metastasis, etc.) as well as potential mechanisms in different types of cancer. However, none of the studies mentioned changes in phenotype of tumor cells or cell morphology caused by high intake of w-6 PUFAs compared to a balanced PUFAs diet (10.1016/j.bmhimx.2016.11.001).
- "Thus, in this work we used an experimental model of pulmonary squamous cell carcinoma to study the aggressiveness of the tumor in mice fed with diets rich in ω-6 97 and its relationship with oxylipins"
Why did you use this kind of " cancer" for your thesis??
Author response a: Lung cancer is the main cause of death from cancer worldwide, as well as has high incidence and low survival rates. Non-small cell lung carcinomas (NSCLC) are the most common subtype of lung cancer in men and women, and squamous cell lung cancer is the second most common histologic subtype, comprising approximately 20% of primary lung neoplasms in the US (10.5334/aogh.2419). Although, it is not so common to associate diet with lung cancer (like colon or gastrointestinal cancer), recent studies have linked consumption of w-6 PUFAs to pro-tumor processes. Those studies focused on analyzing w-6 PUFAs intake as risk factor that might enhances disease progression (10.1093/nutrit/nuab117, 10.3390/ijerph18073700, 10.3390/life12020270).
Additionally, this is a well-known model, and was established by our investigation group because its characteristics allow us to evaluate several alterations in the tissues (10.1038/s41598-020-64146-6).
- The experimental part about the “experimental model of pulmonary squamous cell carcinoma to study the aggressiveness of the tumor in mice fed with diets rich in ω-6 and its relationship with oxylipins2 was now well described, further only the number of references 24-26 are mentioned. So please add detailed description of the work up of the samples.
Author response a: We described the oxylipin profile as the reviewer suggested (Lines 507-516).
- So, please provide the lipidomic profiles of the tissue samples. Further, what is the FAME profile? this is important in order to evidence the remodelling effect of omega 6; please provide a a very nice GC chromatogram picture highlighting each fatty acid recognized. This is important aspect due the connection between the omega 6 (like Arachidonic acid) and oxylipin metabolites. Please perfom the specific statistical correlation between the fatty acid profile, the oxylipin and the signals in order to demonstrate and confirm your thesis.
Author response a: Thank you for your comment. It is a great point to have the lipidomic profiles of the tissue samples as well as the parents-fatty acids profiles. In our previous finding, we found there was a very good correlation of oxylipins in tissue and plasma (10.2131/jts.38.833). Due to the limited resource and with our other work on this topic (10.2131/jts.38.833, 10.1016/j.prostaglandins.2014.07.002, 10.1371/journal.pone.0076575, 10.1016/j.prostaglandins.2014.05.002), we only did the oxylipin profile in plasma.

Round 2
Reviewer 2 Report
Dear Authors
The correlation between fatty acid profiles and oxylipins is a very important aspect because in this work you have examined new matrices
Anyway I suggest you to provide all connections in your future works